# Improvement of Embryo Recovery in Holstein Cows Treated by Intra-Ovarian Platelet Rich Plasma before Superovulation

**DOI:** 10.3390/vetsci7010016

**Published:** 2020-02-01

**Authors:** Fausto Cremonesi, Stefano Bonfanti, Antonella Idda, Lange-Consiglio Anna

**Affiliations:** Dipartimento di Medicina Veterinaria, Università Degli Studi di Milano, 26900 Lodi, Italy; fausto.cremonesi@unimmi.it (F.C.); ste.bonfanti93@virgilio.it (S.B.); antonella.idda@unimi.it (A.I.)

**Keywords:** bovine, superovulation, platelet rich plasma (PRP), follicles, embryo

## Abstract

The current research was designed to evaluate if intra-ovarian administration of autologous platelet rich plasma (PRP) before superovulation could increase the number of follicles responsive to gonadotropin treatment in order to improve embryo recovery in donor cows. Eight Holstein-Friesian cows of proven fertility were employed. After estrous synchronization, at the 18th day of diestrous, the right ovary of each cow was left untreated and served as control while the left ovary was inoculated with 5 mL of PRP. Cows were left to spontaneously return to estrous, and nine days later, a standard superovulation was initiated for every cow. Seven days after artificial insemination (AI), putative embryos were collected by flushing the right and left uterine horns separately. All statistics were calculated by ANOVA. The mean number of follicles, evaluated by transrectal ultrasound scanning, did not statistically differ before PRP treatment between right (control) and left (treated) ovaries (9.18 ± 1.35 and 7.32 ± 1.67, *p* = 0.28, respectively) as well as at 48 h after PRP injection (7.67 ± 2.52 and 8.00 ± 2.00, *p* = 0.73, respectively). A statistical (*p* = 0.023) difference was found in the average number of follicles at the last gonadotropin injection between control and treated ovaries (11.33 ± 2.89 and 20.00 ± 9.17, respectively). The statistically different (*p* = 0.0037) number of grade 1-2 blastocysts harvested from the uterine horn ipsilateral to control ovaries in comparison to that collected from the treated ones (6.63 ± 2.92 and 14.75 ± 5.92, respectively) suggests that intra-ovarian injection of PRP before superovulation could exert beneficial effects both in latent follicle growth and in vivo embryo production.

## 1. Introduction

Currently, the reproductive technology that allows the fastest genetic progress in a herd of cattle is the in vivo production of embryos by superovulation. This reproductive biotechnology is aimed at obtaining as many ovulations as possible from single high genetic merit cows and transferring the embryos obtained into less valuable recipient cows with a high probability of obtaining pregnancies [1]. Superovulatory strategies should aim to stimulate follicular growth or inhibit follicular atresia in order to trigger multiple ovulations from the available supply of ovarian follicles in donor cows. In this context, hormonal or physical treatments have been employed with the main goal of increasing the follicles potentially responsive to superovulation, stimulating their growth from the pool of the ovarian follicular reserves (i.e., reducing the inhibiting action of the dominant versus subordinate follicles) [2].

In the cow, the development of small follicles depends on the high circulating concentrations of follicle stimulating hormone (FSH). With the FSH level declining during the first three days of the follicular wave, most of the follicles stop growing and begin to regress within two to five days of emergence [1,2]. Despite this decline in FSH, the dominant follicle continues to grow, maintaining cellular proliferation and estradiol production [2]. The developmental dynamic of follicles larger than 1 mm is well characterized but that of the smaller follicles remains unknown. The stimulating effect of FSH on the growth of small antral follicles in vitro and in vivo suggest a role for FSH in the development of these follicles [3,4,5]. Then, only if dominant follicles are removed or if exogenous FSH is supplied, subordinate follicles can develop. Basically, in the bovine, this is the mechanism underlying superovulatory protocols that were developed in the 1980s, which were further improved by refining hormonal preparations like human menopausal gonadotropin (HMG), equine chorionic gonadotropin (eCG), or follicle-stimulating hormone (FSH) from porcine or ovine pituitaries [6,7]. However, although many advances have been made in understanding the physiology of superovulation, the treatment does not always lead to optimal results.

It was estimated that every year around the world, approximately half a million bovine embryos are produced and that 70% of these embryos are produced by only 30% of superovulated cows, that a high variability in individual responses exists [7]. The variability of ovarian responses has been correlated to differences in superovulation treatment, such as gonadotropin preparations, lot and dose of gonadotropins, duration and time of treatment and use of supplementary hormones [6]. Further causes for this could also be related to either the animal itself or the environment. These may include nutritional status, reproductive history, age, season, breed, effects of repeated superovulations and ovarian status at the time of treatment [8].

Follicle stimulating and luteinizing (LH) hormones are required for follicologenesis in mammals but there is considerable variability of their content in the preparation of crude gonadotropins. Chupin et al. [9], in a study of bovine superovulation, found that by using a comparable amount of FSH but a different amount of LH, the ovulation rate and the amount of collected embryos increased with decreasing amounts of LH. Furthermore, it has been suggested that high levels of LH during superovulation cause premature activation of the oocyte [10]. Although it is recognized that a certain dose of LH is essential for superovulation success, endogenous LH levels may be suitable. Looney et al. [11] described that recombinant bovine FSH induces high responses to superovulation without the supplementation of exogenous LH. These results propose that LH is not necessary for the bovine superovulation protocol and that the quality of embryos may be higher if only FSH is used [8]. After introduction of prostaglandin F2α (PGF-2α) in the 1970s, superovulatory treatments usually start between day 8 and 12 of the estrus cycle [12,13], corresponding to the period of the second follicular wave emergence in cows that have two or three waves during the cycle [14,15]. Considering the variability of the responses in superovulation and that none of the treatments are able to influence the recruitment process of small ovarian follicle turnover, the aim of this study was to stimulate the growth of small or preantral follicles by administrating platelet rich plasma (PRP) inside the ovary before the superovulation protocol. This hemocomponent is rich in growth factors and cytokines [16] known for their regenerative properties in human [17] and veterinary medicine [18,19]. Referring to human medicine, Sills et al. [20] treated four women with proven infertility for 60 ± 25 months with 5 mL of autologous PRP for each ovary under ultrasound guidance via the transvaginal route. After an in vitro fertilization program, an average of 5.3 ± 1.3 mature oocytes were found at metaphase stage 2, and all patients produced at least one blastocyst suitable for cryopreservation.

The goal of this study was to investigate if administration of autologous PRP inside the bovine ovary before gonadotropin treatment could increase the number of follicles responsive to superovulation in order to improve embryo recovery from eight donor cows.

## 2. Materials and Methods

Permission was obtained from Milan University Bioethics Committee n.119_2017, in accordance with the 2010/63 EU directive on animal protection and Italian Law (D.L. No. 116/1992) and following standard veterinary practice. Written informed consent from the owners was also obtained to allow evaluation of the in vivo effects of PRP in superovulated cows.

### 2.1. Donor Cattle

In this study, eight Holstein-Friesian cows (3 to 4 years of age without any reproductive pathologies) were enrolled. All animals calved normally between 80 and 90 days before the beginning of the experiment. Cows were maintained in one farm, fed with a total mixed ration with free access to water and were showing regular ovarian cycles.

### 2.2. Preparation of PRP by Double Centrifugation

To produce PRP, 250 mL of whole blood from the mammary vein of each cow was collected into blood collection bags containing citrate-phosphate dextrose-adenine (CPDA-1). Whole blood was centrifuged at 100× *g* for 30 min to allow supernatant plasma collection, which then underwent a further centrifugation at 1500× *g* for 10 min. The resulting platelet pellet was diluted with the plasma of each cow to obtain a concentration of 1 × 10^9^ platelets/mL [18,19].

### 2.3. Ultrasound Monitoring of Follicular Waves

After ultrasound examination of the presence of a corpus luteum, all cows were treated with 500 µg of cloprostenol (Estrotek, Fatro, Ozzano dell’Emilia), a synthetic analogue of prostaglandin F2α, in a volume of 2 mL as indicated. This allowed the synchronization of the estrous cycle. Estrus was detected by observing both behavioral signs and the presence of abundant vaginal discharge. Furthermore, the presence of a preovulatory follicle was confirmed by trans-rectal ultrasound. Constant ultrasonography monitoring was performed every other day for two consecutive cycles using an Ibex ultrasound equipped with a linear transrectal 5 megahertz probe to observe follicles with a diameter greater than 3 mm. Data regarding the ultrasound scans were recorded on the ultrasound memory and then transferred to a PC for further analysis. On the 19th day of the second ovarian cycle, monitored as above, cows underwent intra-ovarian PRP injection. After PRP administration, cows were again ultrasound monitored by rectal palpation every other day to register the onset of estrous and the follicular dynamic until the beginning of the superovulation. Thereafter, donor cows were ultrasound monitored during the superovulation treatment.

### 2.4. PRP Administration

In each animal, the right ovary was left untreated and served as control while the left ovary was inoculated with 5 mL of PRP at concentration of 1 × 10^9^ platelets/mL under ultrasound guidance using a Madison SA 600v instrument equipped with an ovum pick up probe (PB-06VE65/20BD). Each animal was placed in a cattle cage and a 5 × 5 cm patch of hair over the S5-C1 vertebrae was shaved. The area was cleaned with alcohol and povidone-iodine three times. Then, a sacrococcygeal epidural anesthesia was induced with 4 mL of 2% procaine hydrochloride (Procamidor, Richter Pharma Ag) and checked by loss of tail tone. Before introducing the ultrasound probe to the vaginal fornix, the vulva was cleaned, and the vagina was lubricated. The ovary was manually directed to the probe by rectal palpation. A spinal 18G needle connected to a steel tube was used as a transvaginal needle was inserted into the probe needle guide and pushed through the fornix to enter the ovarian stroma in order to administer the PRP.

The dose of PRP administered was decided following the procedure of Sills et al. [20] and by tests carried out in our laboratory with bovine ovaries recovered at the slaughterhouse.

### 2.5. Superovulation Treatment

Superovulation was started on the 9th day of the cycle that started after PRP injection and after ultrasound assessment of corpus luteum and ovarian follicular status. Each cow was superovulated using 1000 i.u. of a commercial preparation of gonadotrophin (Gn) (500 i.u. FSH + 500 i.u. LH) (Pluset, Calier, Barcelona, Spain) administered in decreasing doses via intramuscular injection, twice daily for 5 days with an interval of 12 h (first injection at 8 a.m. and the second in the evening) as shown in Table 1.

At the seventh injection, cows also received a luteolytic dose of 500 µg of PGF-2α (Estrotek, Fatro, Ozzano dell’Emilia), to induce estrus. Donor cows were checked for estrus detection at 48h and 60 h after PGF and were inseminated at standing estrus. Inseminations were then repeated 12 h and 24 h later, using one straw of the same frozen sperm of proven fertility.

### 2.6. Flushing to Collect Embryos

To evaluate the effect of PRP injection, the right and left uterine horns were flushed individually seven days after AI. Embryos were collected under epidural anesthesia by Procamidor, Richter Pharma Ag. The uteri were non-surgically flushed using Ringer’s solution (Terumo, Tokyo, Japan) containing 0.1% fetal calf serum (FCS) through a multi-eye 16-French embryo collection catheter. 

Collected embryos were counted and evaluated for quality and stage of development following international embryo transfer society (IETS) criteria [21].

### 2.7. Statistical Analyses

Data were analyzed with RStudio Version 1.2.1335. Data normality was evaluated with the Shapiro–Wilk normality test. For non-normally distributed data, the Kruskal–Wallis non-parametric test was used. Differences were considered statistically significant at *p* < 0.05.

## 3. Results

### 3.1. Ultrasound Monitoring of Follicular Waves

The follicular waves were monitored by ultrasound before and after PRP administration and during superovulation in control ovaries (Figure 1) and in treated ovaries (Figure 2) at the same time point.

The number of follicles before PRP treatment, two days post-treatment with PRP and after superovulation, was detected by ultrasound examination. Data are summarized in Table 2. 

No statistically significant differences (*p* = 0.28) were detected between treated and control ovaries during the pre-treatment step or two days post-treatment with PRP. After the superovulation protocol, a statistical significant difference (*p* = 0.023) was detected between treated and control ovaries.

Inside the treated ovaries, a statistically significant difference was detected between pre-treatment with PRP and after superovulation (*p* = 0.0018).

### 3.2. Flushing and Collection of Embryos

By non-surgical flushing, 6.63 ± 2.92 grade 1–2 blastocysts were harvested from the uterine horn ipsilateral to the control ovaries, while 14.75 ± 5.92 blastocysts were flushed from the treated ones (*p* = 0.0037). 

In Table 3, the number of collected oocytes, morulas and blastocysts from both ovaries is reported.

## 4. Discussion

To date, in vitro embryo production techniques produce embryos of poor quality compared to those collected in vivo in both human and veterinary medicine [22,23]. Looking towards innovations in the production of bovine embryos, in vivo production remains the way forward. The protocol requires a superovulation treatment that is able to let subordinated follicles grow into ovulatory follicles despite the action of the dominant follicle. This reproductive biotechnology is used widely around the world but, independently from the type of gonadotropin used, the variability of the individual response is very high.

Indeed, only 30% of superovulated bovines provide most of the in vivo embryos produced [7]. Given this variability, the ovarian response could be enhanced using substances rich in growth factors or cytokines that could improve the ovarian microenvironment and the recruitment of small follicles.

The use of PRP in the bovine reproductive tract was first suggested by Lange-Consiglio et al. [19] and in the human ovary by Pantos et al. [24] and Sills et al. [20]. Inside their α-granules, platelets contain growth factors, chemokines and cytokines [25] that act in a paracrine manner on cell migration and proliferation and on matrix synthesis [16,17]. Transforming growth factor β1 (TGF-β1) and TGF-β2, platelet derived growth factors (PDGF-AA, PDGF-BB, PDGF-AB), stromal derived growth factor-1 (SDF-1), fibroblast growth factor (FGF), insulin-like growth factor 1 (IGF-I), vascular endothelial growth factor (VEGF), epidermal growth factor (EGF), and hepatocyte growth factor (HGF) are very important for regeneration processes. Indeed, these growth factors act synergistically to control recruitment, proliferation and activation of fibroblasts, neutrophils, monocytes and macrophages to promote angiogenesis, fibroplasia, matrix deposition and re-epithelialization in order to induce tissue regeneration [17]. Lange-Consiglio et al. [19] used PRP to treat repeat breeder cows that were not able to get pregnant because of fertilization failure or early embryonic death. Intrauterine administration of PRP 48 h after AI improved the pregnancy rate of treated cows compared to control animals. These results suggest that PRP was able to improve the uterine environment and restore the endometrial-embryo ‘dialogue’ to guarantee implantation and subsequent pregnancy. In the Pantos study [24], a group of eight infertile women were treated by intraovarian injection of PRP followed by in vitro fertilization (IVF). The result was the birth of healthy individuals after transfer of frozen-thawed embryos derived from non-donor oocytes. Similarly, Sills et al. [20] treated four women with proven infertility by intra-ovarian PRP administration and, after an in vitro fertilization program, all patients produced at least one blastocyst suitable for cryopreservation.

In the cow, our results show that after the superovulation protocol, a statistically significant difference (*p* ≤ 0.05) was detected between PRP treated and control ovaries with an average of 20.00 ± 9.17 follicles in treated ovaries compared to 11.33 ± 2.89 in the control. In addition, inside the treated ovaries, a statistically significant difference was detected between pre-treatment with PRP and after superovulation (7.32 ± 1.67 vs 20.00 ± 9.17; *p* ≤ 0.05), while no differences were detected in the control ovaries. After AI, 6.63 ± 2.92 grade 1–2 blastocysts were collected by flushing from the uterine horn ipsilateral to control ovaries and 14.75 ± 5.92 from the treated ones, highlighting once again the statistically different result (*p* < 0.05) of the treatment compared to the control. 

The results of Pantos et al. [24], Sills et al. [20] and our work indicate that PRP has a very important impact on follicular recruitment, although there are some differences between our study and those just cited [20,24]. At first, we studied healthy cows enrolled as donor animals to obtain in vivo embryos while, in the previously cited papers, women of proven infertility were treated. The results were promising in every case.

Many hypotheses can be raised about the mechanisms of action of PRP. Inside the ovary, PRP could distribute mediators regulating angiogenesis, and then ovarian perfusion, with the aim of improving the competence of the oocyte [20]. It can also be assumed that a balance is established between apoptosis and cellular survival due to the presence of pro-apoptotic factors [i.e., Fas-Ligand, CD40L, tumor necrosis factor-related apoptosis inducing ligand (TRAIL), tumor necrosis factor (TNF)-like weak inducer of apoptosis (TWEAK), and tumor necrosis factor superfamily member 14 (TNFSF14 or LIGHT)] and anti-apoptotic factors (i.e., epatocyte growth factor, stromal-derived factor-1, serotonin, adenosine diphosphate, and sphingosine- 1-phosphate) [26]. Another hypothesis is that the vast range of cellular signals contained in PRP acts by favoring the recruitment of latent follicles simply by awakening them with the arrival of numerous growth factors in combination with the gonadotropin treatment. Finally, it can be assumed that growth factors are able to communicate with ovarian stem cells to induce their differentiation in ex novo oocytes [20]. This last hypothesis predicts post-natal oogenesis in the adult ovary, a controversial concept that has led to universally accepting that the ovary, at birth, is equipped with a predefined pool of oocytes that, with ageing, undergoes an irreversible decline. Considering that a niche of stem cells is present in each type of tissue, this hypothesis may not be so fanciful considering that there are already papers and protocols aiming to isolate and identify ovarian stem cells [27,28,29].

Regardless of the different hypotheses, our results indicate that intra-ovarian treatment with PRP has a very important impact on follicular recruitment.

Further studies will be needed to clarify in detail the mechanisms by which platelet growth factor acts at the ovarian level, especially by expanding the number of enrolled animals. Nonetheless, it can be suggested that this technique has the potential to deeply innovate the embryo production procedure in vivo.

The proposed methodology is very innovative and without adverse effects. Blood sampling is part of normal routine breeding practice. In addition, the blood for the production of the platelet lysate comes from and is used on the same animal from which it is taken, making this treatment absolutely autologous (collection and inoculation in the same animal). In this way, this treatment is free from risks of transmission of viral or bacterial agents between animals. The PRP preparation is inexpensive and very simple (based on two centrifugations) but a sterile laboratory is required. The intra-ovarian administration of PRP may be difficult, but a veterinarian experienced in the reproductive field and able to perform ultrasound and ovum pick-up should have no problem in proceeding with this technique.

Although the precise mechanism of action of PRP is unknown, these results show that the use of intra-ovarian PRP improves the recruitment of small follicle in a program of superovulation on healthy cows without reproductive issues. In addition, considering the improvement of the ovarian microenvironment, PRP could have important applications even in fields other than superovulation. In fact, one of the major problems on dairy farms is poor reproductive indices in the summer period which is characterized by a low conception rate due to the heat stress which compromises the quality of the oocyte. Intra-ovarian PRP treatment before this season could improve the quality of the oocyte.

A further possible application could be to use the PRP on high-production dairy cows that do not manifest estrus after the 90th day of lactation. In this case, it would be interesting to see if PRP is able to stimulate follicologenesis in order to restart ovarian activity. It is remarkable to note that in this case, cows would be comparable to the human patients of Sills et al. [20].

## 5. Conclusions

Our data suggest that PRP could stimulate latent follicles and in vivo embryo production. In future perspectives, in addition to increasing of the number of animals, we hope to be able to investigate both the specific mechanisms by which platelet factors influence follicular recruitment and the possible further applications of PRP in the reproductive field.

## Figures and Tables

**Figure 1 vetsci-07-00016-f001:**
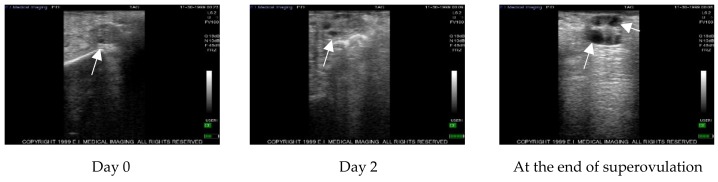
Example of echographic images representing the change in follicle number during the different steps of the study in control ovaries. Arrows indicate follicles.

**Figure 2 vetsci-07-00016-f002:**
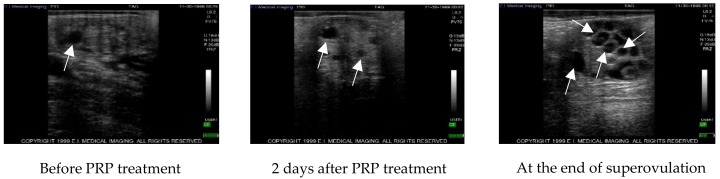
Example of echographic images representing the change in follicle number during the different steps of the study in treated ovaries. Arrows indicate follicles. PRP: platelet rich plasma.

**Table 1 vetsci-07-00016-t001:** Scheme of superovulation treatment. Gn: gonadotrophin; *: seventh injection.

Day of Treatment	First Gn Injection (i.u)	Second Gn Injection (i.u.)
1	150	150
2	125	125
3	100	100
4	75*	75
5	50	50

**Table 2 vetsci-07-00016-t002:** Number of follicles detected at different steps in treated or control ovaries.

Steps	Average Number of Follicles Treated Ovary	Average Number of Follicles Control Ovary
Pre-treatment with PRP	7.32 ± 1.67 ^aA^	9.18 ± 1.35 ^aA^
Two days post-treatment with PRP	8.00 ± 2.00 ^aA^	7.67 ± 2.52 ^aA^
After superovulation	20.00 ± 9.17 ^aB^	11.33 ± 2.89 ^bA^

Value are express as mean ± standard deviation (SD). Within lines, values with different superscript letters (a,b) differ significantly (*p* ≤ 0.05). Within columns, values with different capital letters (A,B) differ significantly (*p* ≤ 0.05).

**Table 3 vetsci-07-00016-t003:** Results of flushing from both ovaries in each cow.

Cows	Degenerated or Not Fertilized Oocytes		Morulas		Blastocysts	
	Control	Treated	Control	Treated	Control	Treated
1	1	4	2	5	5	16
2	1	0	2	1	1	6
3	0	0	2	0	6	10
4	1	0	0	2	8	18
5	0	1	1	1	11	17
6	0	0	2	3	8	24
7	0	1	0	1	8	9
8	0	1	0	1	6	18
TOT	0.38 ± 0.52 ^a^	0.88 ± 1.36 ^a^	1.13 ± 0.99 ^a^	1.75 ± 1.58 ^a^	14.75 ± 5.92 ^a^	6.63 ± 2.92 ^b^

Value are express as mean ± standard deviation (SD). Within lines, values with different superscript letters (a,b) differ significantly (*p* ≤ 0.05).

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
