# Peer review of "Improvement of Embryo Recovery in Holstein Cows Treated by Intra-Ovarian Platelet Rich Plasma before Superovulation"

_vetsci, 2020, doi:10.3390/vetsci7010016_

Round 1

Reviewer 1 Report

Please, find the attached comments,

Abstract,

Line 13; PGF2alfa >>>PGF2α

Line 18,20; (P≥0.05) >>> please, mention the value (e.g. P = 0.67).

Line 23; Our data suggest that PRP >>>Our data suggest that intra-ovarian PRP

Introduction,

Many sentences are messing References citation,

Materials and Methods,

Line 164-166; Please, mention the the post-hoc test.

Results,

Figure 1; the authors added echographic figures of the PRP treated ovary. It will be more informative if you can add the control (untreated Ovary) parelle to this.

Author Response

The Authors thanks the Reviewers and the Editor for considerations and helpful suggestions. According to the comments and suggestions, we have carefully evaluated all critical points and the manuscript has been thoroughly revised. The Authors hope that now the manuscript is suitable for publication on “Veterinary Science”.

The paper has been revised for plagiarism but occasionally some sentence related to composition of PRP or its protocol preparation could not be changed. For other sentences, some changes have been made and some reference was inserted.   

REVIEW 1

Abstract,

Line 13; PGF2alfa >>>PGF2α

ANSWER: the authors thank the referee for her/his suggestion. The PGF2 alfa has been corrected

Line 18,20; (P≥0.05) >>> please, mention the value (e.g. P = 0.67).

ANSWER: the authors thank the referee for her/his suggestion.The values have been added

Line 23; Our data suggest that PRP >>>Our data suggest that intra-ovarian PRP

ANSWER: the authors thank the referee for her/his suggestion. The sentence has been modified

Introduction,

Many sentences are messing References citation,

ANSWER: the authors thank the referee for her/his suggestion. Some citations in the first sentences of the Introduction have been added

Materials and Methods,

Line 164-166; Please, mention the the post-hoc test.

ANSWER: the authors thank the referee for her/his suggestion. The sentence has been modified

Results,

Figure 1; the authors added echographic figures of the PRP treated ovary. It will be more informative if you can add the control (untreated Ovary) parelle to this.

ANSWER: the authors thank the referee for her/his suggestion. A new figure (1) has been added.

We feel that we have addressed all of the queries raised by the referees and Editorial board member and  we hope that the paper is now acceptable for publication in Veterinary Science.

We thank you in advance for your time and consideration.

On behalf of all authors best regards,

Anna Lange Consiglio

Reviewer 2 Report

Minor changes:

Line 22: Change the Italian word omolateral by the English ipsilateral here and in all the paper (Lines 211, 254, etc.).

Line 39: Define the acronym FSH in this line (in the paper appears by first time in line 50).

Line 62: Define the acronym LH in this line.

Line 96: Indicate the number of calvings of cows.

Line 103: Define the acronym CPDA-1.

Line 115: Separate -an Ibex-.

Line 141: Add the Pluset composition after 1000 u.i.: (500 u.i. FSH + 500 u.i. Lh).

Line 148: Write scheme with a capital letter: Scheme.

Line 189: Unify in all the paper after Table # : or .

Mayor changes:

Results

Line 212: Describe in a table, by each grade, the number of collected blastocysts of both ovaries.

Discussion

Line 293: Discuss possible objections, adverse events, difficulties, etc. of this technic. For example ovary trauma, selection of character undesirable in the progeny, etc.

Author Response

The Authors thanks the Reviewers and the Editor for considerations and helpful suggestions. According to the comments and suggestions, we have carefully evaluated all critical points and the manuscript has been thoroughly revised. The Authors hope that now the manuscript is suitable for publication on “Veterinary Science”.

The paper has been revised for plagiarism but occasionally some sentence related to composition of PRP or its protocol preparation could not be changed. For other sentences, some changes have been made and some reference was inserted.   

REVIEW 2

The authors thank the referee for her/his suggestions.

Minor changes:

Line 22: Change the Italian word omolateral by the English ipsilateral here and in all the paper (Lines 211, 254, etc.).

Line 39: Define the acronym FSH in this line (in the paper appears by first time in line 50).

Line 62: Define the acronym LH in this line.

Line 96: Indicate the number of calvings of cows.

Line 103: Define the acronym CPDA-1.

Line 115: Separate -an Ibex-.

Line 141: Add the Pluset composition after 1000 u.i.: (500 u.i. FSH + 500 u.i. Lh).

Line 148: Write scheme with a capital letter: Scheme.

Line 189: Unify in all the paper after Table # : or .

ANSWER: All the minor changes has been done

Mayor changes:

Results

Line 212: Describe in a table, by each grade, the number of collected blastocysts of both ovaries.

ANSWER: the authors thank the referee for her/his suggestion. A new table (table 3) has been added.

Discussion

Line 293: Discuss possible objections, adverse events, difficulties, etc. of this technic.

ANSWER: the authors thank the referee for her/his suggestion. A new sentence has been added from line 322 to line 328.

We feel that we have addressed all of the queries raised by the referees and Editorial board member and  we hope that the paper is now acceptable for publication in Veterinary Science.

We thank you in advance for your time and consideration.

On behalf of all authors best regards,

Anna Lange Consiglio

Round 2

Reviewer 1 Report

All comments have been addressed in the revised manuscript 

Author Response

Thank you.